# Innate Immune Response Assessment in *Cyprinus carpio* L. upon Experimental Administration with *Artemia salina* Bio-Encapsulated *Aeromonas hydrophila* Bacterin

**DOI:** 10.3390/vaccines11040877

**Published:** 2023-04-21

**Authors:** Akshaya Radhakrishnan, D. S. Prabakaran, Thiyagarajan Ramesh, Ramalingam Sakthivel, Kavikumar Ramasamy, Hyo-Shim Han, Sivakamavalli Jeyachandran

**Affiliations:** 1Department of Biotechnology & Microbiology, National College (Autonomous), Trichy 620001, Tamil Nadu, India; 2Department of Radiation Oncology, College of Medicine, Chungbuk National University, Cheongju 28644, Republic of Korea; 3Department of Biotechnology, Ayya Nadar Janaki Ammal College (Autonomous), Sivakasi 626124, Tamil Nadu, India; 4Department of Basic Medical Sciences, College of Medicine, Prince Sattam bin Abdulaziz University, P.O. Box 173, Al-Kharj 11942, Saudi Arabia; 5School of Electrical Engineering, Chungbuk National University, Cheongju 28644, Republic of Korea; 6Department of Biotechnology, Sunchon National University, Suncheon 57922, Republic of Korea; 7Lab in Biotechnology & Biosignal Transduction, Department of Orthodontics, Saveetha Dental College & Hospitals, Chennai 600077, Tamil Nadu, India

**Keywords:** *Cyprinus carpio* L., *Aeromonas hydrophila*, Motile Aeromonas Septicemia, inactivated vaccine, innate immunity, bio-encapsulation, *Artemia salina*

## Abstract

The present study aimed to analyze the enhancement of innate immune responses in juvenile-stage common carp (*Cyprinus carpio* L.), upon the administration of heat-killed *Aeromonas hydrophila* at a dosage of 1 × 10^7^ CFU ml^−1^ through bio-encapsulation in the aquatic crustacean, *Artemia salina*. This work emphasizes the modulation of innate immune response when administered with the bio-encapsulated heat-killed antigen that acts as an inactivated vaccine against Motile Aeromonas Septicemia disease. Bio-encapsulated oral administration of antigens promotes innate immunity in juvenile-stage fishes. The optimization of effective bio-encapsulation of bacterin in *Artemia salina* nauplii was carried out and the best optimal conditions were chosen for immunization. The functional immune parameters such as myeloperoxidase, lysozyme, alkaline phosphatase, antiprotease and respiratory burst activity in serum, blood and intestinal tissue samples were analyzed along with blood differential leukocyte count and tissue histopathology studies. Both humoral and cellular immune responses analyzed were substantially induced or enhanced in the treatment groups in comparison with the control group. The results showed a significant variation in the bio-encapsulation group than the control group and also were comparable to the protection conferred with immersion route immunization under similar conditions. Thus, most of the innate non-specific immune responses are inducible, despite being constitutive of the fish immune system, to exhibit a basal level of protection and a road to better vaccination strategy in *Cyprinus carpio* L. aquaculture worldwide.

## 1. Introduction

Aquaculture is regarded as the fastest-growing food production sector globally. Due to the excessive density, low dissolved oxygen (DO) levels, sub-optimal water quality, and high pathogen loads, the culture environment whether in tanks, cages, lakes or ponds, seawater, or freshwater, might stress the fish to a greater extent [1]. The interactions of pathogen, host and corresponding environment are to be well acknowledged to analyze the emergence and progression of fish disease and its associated factors. Interdisciplinary studies involving analysis of the characteristics of potentially pathogenic microorganisms in fish, biological aspects of the fish and its hosts, and a better understanding of the environmental factors affecting them will allow adequate prevention measures to be implemented to prevent and control the major diseases limiting the production of freshwater and marine fishes [2].

Vaccination is also the method of inducing protective immune responses in fish against infectious pathogens by exposing them to non-pathogenic forms or components of microorganisms. Easier, feasible, cost-effective and safe inactivated vaccines, considering the risks associated with live attenuated vaccines, are essential for controlling infectious disease on a large scale. The effectiveness of vaccines is greatly affected by various factors including age, dosage, route of administration, handling stress and clear understanding of the mechanism of the immune response towards the antigen. Various routes of administration of vaccines also play a major role in the immune response in fish [3]. Most common vaccination routes include intra-peritoneal injection and immersion of inactivated antigen which induces a stronger innate immune defence but are associated with certain disadvantages such as handling and post-injection stress, not being applicable in large-scale farms and less protection in terms of immersion route.

Oral uptake through feed seems to be effective in terms of easier handling with reduced stress, cost effectiveness and applicability on a larger scale in comparison to injection and immersion route vaccines. Oral vaccines through formulated feeds are subjected to degradation by digestive enzymes and this can be overcome by their bio-encapsulation through live feeds such as nauplii of *Artemia salina*, which also promotes digestive immunity [4]. Effective innate immune response through inactivated antigens, produced by low-cost manufacturing methods, provides a great incentive to resume and perform research on oral vaccines and production in aquaculture. Understanding the structure and function of the fish immune system upon exposure to such inactivated antigenic components is critical for the development of new technologies and products to increase production [5]. It would be fascinating to investigate the modulation of various innate immune parameters as well as the amount of augmentation following the administration of effective therapeutics/immune modulators.

In that aspect, the present study carried out the assessment of the innate immune response of *Cyprinus carpio* L. (common carp), an economically significant freshwater fish (US$ 3–5 per kg), upon the administration of heat-inactivated *Aeromonas hydrophila* which is the causative agent of Motile Aeromonas Septicemia (MAS) disease in large-scale fish cultivation across the world [6]. Hence, to unveil the significance of bio-encapsulated antigenic delivery in common carp against *Aeromonas hydrophila* infection, which is not reported yet, this study is focused primarily on the innate immune analysis upon antigenic administration through bio-encapsulated *Artemia salina*. Thereby, it aims in assessing the role of this antigen to function as an inactivated vaccine against MAS through an enhanced route and dosage of administration for a strong immune response in common carp that could be commercialized.

## 2. Materials and Methods

### 2.1. Experimental Animals

We used a total of 90 (15.4 ± 0.15 mm) juvenile *Cyprinuscarpio* L. obtained from a local fish hatchery at Tanjavur, Tamil Nadu, India. The fish were held in 6 fibreglass tanks (15 fish per tank) and initially fed with a commercial diet at a ration of 2% of their body mass twice a day. Tanks were filled with water at 30° C, and the oxygen and ammonia levels were monitored daily (6.8 mg L^−1^ and 0.35–0.37 mg L^−1^, respectively) with repeated water exchange for the removal of sludge. The photoperiod was 14 h of light and 10 h of darkness [7].

### 2.2. Experimental Design

After 4 weeks of acclimation, the fish were transferred to three experimental tanks (15 fishes (15.4 ± 0.15 mm) per tank). All experiments were conducted in duplicates. The three experimental groups were a bio-encapsulated antigen-fed group (B), an immersion antigen group (I) and a control group (C). The (B) group were fed with the *A. hydrophila* antigen bio-encapsulated Artemia nauplii immediately after the bio-encapsulation process (given below) on the first three days of the experiment. The (I) group fishes were immersed in the prepared antigen (given below) in a 2 L tub for the 30 s on the first day of the experiment. The (C) group fishes were fed with artemia nauplii incubated in sterile PBS.

After 30 days of the experiment, a similar booster dosage of antigen was administered to all of the fishes in (I and B) groups (control group fed with hatched artemia) in a similar manner. Fishes from all groups were sampled on days 0, 7, 14, 30 a 60 during the experiment (n = 2). We measured serum and tissue non-specific immune parameters such as lysozyme, myeloperoxidase, respiratory burst activity and antiprotease activity along with differential leukocyte count and tissue histological study [8,9].

### 2.3. Preparation of Inactivated A. hydrophila

The *Aeromonas hydrophila* pure culture was obtained from the Microbial Type Culture Collection (MTCC-12301). The bacteria were stored in BHI (Brain Heart Infusion Broth, HiMedia, Kennett Square, PA, USA) medium with 30% glycerol (sterile) at −80 °C. For the experiment, an aliquot of 20 µL was added to 5 mL of autoclaved BHI medium and cultured in a bacteriological incubator at 90 rpm for 24 h at 30 °C. Following that, 100 mL of autoclaved BHI medium was added and incubated under the same conditions. This bacterial culture was centrifuged for 20 min at 12,000× *g*, and the supernatant was discarded. Following that, the residual pellet was washed twice with PBS buffer (0.01 M) and centrifuged for 20 min at 12,000× *g*. Finally, the pellet was resuspended in PBS (0.01 M) and adjusted according to the McFarland turbidity standard to generate a suspension of 1 × 10^7^ CFU mL^−1^, which was inactivated for 30 min in a water bath at 100 °C (optimized). Inactivation was verified by streaking the culture on nutrient agar plates and incubating it at 37 °C for 48 h. The plates with no colonies were confirmed to be inactivated in comparison to the control plates. The preparation was stored at 4 °C [10].

### 2.4. Preparation of Bio-Encapsulated Antigen

#### 2.4.1. Hatching of Cysts

An amount of 0.5 g of *Artemia salina* cysts (procured from commercial Megha farms, Kerala, India) were washed and dissolved in 500 mL of (34 ppt–17 g of sea salt) salt water in a 1 L beaker and allowed to hatch for 48 h with continuous aeration and light. The hatched artemia was observed under light microscopy for the instar II stage meta-nauplii. The artemia at the proper stage was collected and washed prior to the bio-encapsulation process [11].

#### 2.4.2. Optimization of Bio-Encapsulation

The maximum uptake of bacterin by artemia over a period of incubation was optimized by incubating the artemia at a density of 100 nauplii mL^−1^ in a 6-well plate containing 10^7^ CFU mL^−1^ bacterin suspension. Bacterin suspension and nauplii in sterile saltwater separately were used as controls. Each treatment was executed in duplicate. The bacterin was coated in Tryphan blue dye (0.4%) to facilitate the microscopic observation of ingestion by artemia. The uptake of bacterin in artemia gut was observed in a light microscope at different time intervals such as 0, 15, 30, 45, 60, 90, 120 and 180 min, respectively. The absorbance of the suspension at OD_600_ at the same intervals of time was measured, respectively [12].

To analyze the retention capacity of nauplii, half of the nauplii were removed from the suspension, thoroughly washed, and transferred to a flask containing sterile seawater. The absorbance of the suspension at OD_600_ at intervals of time 0, 15, 30, 45, 60, 90, 120, 180 and 240 min was measured, respectively. To determine the concentration of bacteria uptake and evacuation in artemia, a standard concentration curve was prepared from the *Aeromonas hydrophila* isolate by using the plate count technique [13] (Figure 1).

#### 2.4.3. Bio-Encapsulation

Bio-encapsulation of heat-killed *A. hydrophila* was carried out at (n = 100) artemia with 10^7^ CFU mL^−1^ bacterin suspension for 2 h in a 100 mL beaker. The suspension was aerated continuously for optimal conditions. The uptake was confirmed with gut observation under a light microscope. The control group artemia was incubated with PBS under the same conditions for each group. The bio-encapsulated artemia was filtered and washed in sterile water and immediately fed to the animals for immunization or stored in PBS at 4 °C until further use [14,15].

### 2.5. Preparation of Antigen for Immersion

The inactivated *Aeromonas hydrophila* cells in broth were collected by centrifugation at 5000 rpm for 15 min and washed three times with sterile phosphate buffer solution (PBS, 0.1 M, pH 7.0). The cells were re-suspended in sterile PBS and the concentration of cells was adjusted to 1.0 × 10^8^. CFU mL^−1^. The heat-inactivated vaccine was diluted to a ratio of 1:10 to give 1 L of vaccine at 1 × 10^7^ CFU mL^−1^ and was utilized for immersion immunization [14,15].

### 2.6. Fish Sampling

Fishes were anaesthetized and utilized for the collection of blood, serum and tissue samples. Blood was drawn from the anterior cardinal vein of fish from each tank with a 26-gauge needle linked to a 1 mL sterile glass tuberculin syringe. After allowing the blood samples to clot overnight at 4 °C in tubes, the serum was separated by centrifugation at 400× *g* for 10 min and kept at −20 °C until utilized in assays. The blood samples were also collected in separate tubes with anticoagulants (10% EDTA). The intestinal tissue samples were collected through dissection and stored in PBS (PBS, 0.1 M, pH 7.4). Later, the tissues were homogenized in PBS and centrifuged at 10,000 rpm for 10 min. The supernatant was transferred to clean labelled tubes and stored at −80 °C for enzymatic analysis [16,17].

### 2.7. Differential Leucocyte Count of Blood Sample (DLC)

Leishman’s staining was used to perform a differential leucocyte count for all experimental groups. Methanol-fixed blood smears were made immediately after collecting the blood from each group of fish at 14 d.p.v and stained with Leishman’s stain according to the standard method [18].

### 2.8. Non-Specific Immune Parameters

#### 2.8.1. Lysozyme

The serum and intestinal tissue lysozyme activity were measured according to the protocol of [19]. An amount of 150 μL *Micrococcus lysodeikticus* culture (NCT 24, obtained from National College Culture Collection Centre, Tiruchirapalli, TamilNadu, India—www.ncccc.in) at a concentration of 0.2 mg/mL (in 0.04 M PBS, pH-6.2) was added to 15 μL serum in a 96-well U-bottom microtiter plate, immediately vortexed, and the OD_450_ (UV absorbance value at 450 nm) (A0) was measured at 0.5 and 4.5 min. Each test was carried out in triplicate. One lysozyme activity unit (U) was defined as the quantity of serum lysozyme that generated a 0.001 decrease in absorbance at 450 nm per minute.

#### 2.8.2. Myeloperoxidase

Ref. [20] approach was used to assess the serum and intestinal tissue myeloperoxidase activity. An amount of 15 µL of serum was diluted in 135 µL of Hank’s balanced salt solution (free of Ca^2+^ and Mg^2+^), and 50 µL of 20 mM 3, 3′, 5, 5′ tetramethylbenzidine (TMB) (SRL chemicals) and 5 mM H_2_O_2_ were added. After 2 min, the reaction was halted by adding 50 µL of 4 M H_2_SO_4_, and the optical density (OD) was measured in a spectrophotometer at 450 nm.

#### 2.8.3. Respiratory Burst Activity

A decrease in nitroblue tetrazolium (NBT) by the intracellular superoxide radicals was used to quantify respiratory burst activity. An amount of 100 µL of heparinized blood from each group of fish was mixed in 100 µL of 0.2 percent NBT (Hi-media) solution. Following incubation, 50 µL of this mixture was added with 1 mL of N.N. diethylmethyl formamide and centrifuged at 3000× *g* for 5 min, and the optical density (OD) was measured in a spectrophotometer at 540 nm [21].

#### 2.8.4. Serum Antiprotease Activity

The total antiprotease activity of the serum samples was measured using a modified version of [22]. An amount of 10 µL of test serum was mixed with 100 µL of trypsin (bovine pancreas type I, Sigma-Aldrich, St. Louis, MO, USA) and incubated at 25 °C for 30 min with 110 µL PBS as a negative control, 10 µL PBS mixed with 100 µL of trypsin as a positive control, and 10 µL of test serum mixed with 100 µL PBS as a serum blank. The samples were then incubated for 15 min at 25 °C with 1 mL of casein (Sigma-Aldrich, St. Louis, MO, USA) dissolved in PBS (pH 7.2) at 2.5 mg per ml. The reaction was stopped with the addition of 500 µL of 10%trichloroacetic acid (TCA). The samples were centrifuged at 10,000× *g* for 10 min. The OD of the supernatant was measured at 580 nm. The percentage inhibition of trypsin activity was estimated by comparing the absorbance reading average to the value for 100% enzyme activity.
% trypsin inhibition = % trypsin OD − test sample OD/% trypsin OD × 100

#### 2.8.5. Alkaline Phosphatase

Ref. [23] method was used to measure the alkaline phosphatase activity of the intestinal tissue samples. The test solution was a 1:1 combination of diethanolamine HCl buffer (pH 9.8) and magnesium chloride, with p-nitrophenyl phosphate as the substrate. A sample of 20 μL of tissue supernatant was obtained, and 1 mL of the reaction solution was added to it. The sample and reaction solution were properly mixed and allowed to stand for 1 min. The absorbance of the supernatant was then measured in a spectrophotometer at 405 nm over a 1–3 min reaction period. The calibration of the blank was accomplished by storing distilled water in the blank. The samples were tested in triplicate, while the blanks were tested in duplicate.

#### 2.8.6. Histology

The intestine samples of all three groups (I, B and C) on 1 d.p.v and day after the pathogenic challenge were isolated after dissection and stored in 10% neutral buffered formalin. Histology was performed on 5 μm paraffin-embedded tissue slides on poly-L-lysine coated glass slides using the method of [24]. The slides were examined under a microscope after being rinsed with TBS containing 0.05 % Tween 20.

### 2.9. Challenge with Live Aeromonas hydrophila

All fish were starved for 24 h prior to the challenge. Challenge was performed by intra-peritoneal injection with 100 µL virulent strain of *A. hydrophila* culture (10^7^ CFU mL^−1^) suspended in PBS to all group fishes with a syringe after 60 days post-experiment [25]. The fish were observed two times a day for changes in behavioral and clinical conditions. The fish were observed for 2 weeks after the challenge for mortality, lesions development and any signs of abdominal dropsy. The Amend formula was used to calculate the relative percent survival (RPS) [26].

### 2.10. Statistical Analyses

For group comparisons, the experiments’ mean values and standard deviations were utilized. All data were subjected to ANOVA, and the *t*-test in Origin Pro 2022 software (10.0.0.154) was utilized to perform group comparisons to evaluate if there was a statistically significant difference in immunological parameters between each group and the control. At *p* ≤ 0.05, all computations were considered to be significant. Data are provided as means ± standard deviations [27].

## 3. Results

### 3.1. Bio-Encapsulation

Heat inactivation was performed at various temperatures in relation to time and was optimized based on complete inactivation on plates. Complete inactivation was achieved at 100 °C for 30 min. Inactivation was confirmed with no growth colonies on nutrient agar plates. After the incubation of artemia in bacterin suspension, over a period of 4 h, maximum intake was observed at 120 min. The absorbance of the suspension media was recorded at every 30 min interval and a decrease in OD_600_ value from 1.0 to 0.2 was observed at 120 min, indicating the uptake of bacterin by artemia. After 120 min, the OD_600_ value showed a constant trend value indicating neither increase nor decrease and was taken as the optimum period of incubation with the incubation density of 200 nauplii per ml of bacterin suspension at 10^7^ CFU mL^−1^ (Figure 2a).

During bacterin retention analysis of the bio-encapsulated nauplii in sterile PBS, the OD_600_ showed a constant value over 90 min after which a slight increase in absorbance was seen around 120 min with a steady maximum increase from 0 to 0.2 OD over 8 h. This indicates the complete release of bacterin from the artemia gut. Maximum retention of bacterin was observed between 0 min to 120 min after the encapsulation process (Figure 2b). The microscopic images of the Artemia gut also revealed the complete uptake and retention of tryphan blue-coated bacterin over the gut (Figure 3).

### 3.2. Differential Leucocyte Count (DLC)

Differential leucocyte count (mean ± SD percentage) of the blood samples obtained from *Cyprinus carpio* L. immunized with inactivated *A. hydrophila* at 14 days post-immunization showed no significant difference between (*p* ≥ 0.05) the percentage of lymphocytes, monocytes and different granulocytes in all three groups (C, B and I). The percentage of variation of lymphocytes was 85 ± 1.0 in control to 82.6 ± 1.15 in immersion and 82.6 ± 2.08 in bio-encapsulation. Monocytes varied from 4.6 ± 0.5 to 4.6 ± 1.15 and 3.6 ± 1.5 and granulocytes ranged from 11 ± 1.0 to 12.6 ± 2.0 and 12 ± 2.6 among the control and bio-encapsulated groups, respectively, as given in Table 1.

### 3.3. Non-Specific Immune Parameters

#### 3.3.1. Myeloperoxidase

Serum myeloperoxidase (MPO) activity was significantly (*p* ≤ 0·05) enhanced in both immersion and bio-encapsulated groups in comparison with that of control on days 7, 32 and 50. There was also a significant difference (*p* ≥ 0·05) between the immersion and bio-encapsulated groups which showed increased MPO activity among the immersion route at day 50. The bio-encapsulation route exhibits higher and more constant MPO activity than the control group and persisted over day 14, which increased by day 32 after the booster dose. A decrease in the MPO activity from day 7 to day 14 was also seen in the immersion route, which was enhanced by the booster dose. The highest activity was observed at day 50 by 0.4 OD in the immersion route (Figure 4).

#### 3.3.2. Lysozyme

The serum lysozyme activity of *Cyprinus carpio* L. showed enhanced activity of 3985 mg/mL at day 7 in the bio-encapsulated group that was stable over days 15, 32 and 50 with a booster dose at day 30. Increased lysozyme activity was seen at day 50 by 4560 mg/mL and was not significantly different to the immersion group, which showed higher activity than both groups. Both the immunization groups were significantly different from the control group. The booster dose enhanced the lysozyme activity among the immersion group after day 32 (Figure 5).

#### 3.3.3. Respiratory Burst Activity (RBA)

The respiratory burst activity (RBA) of *Cyprinus carpio* L. was significantly less (*p* ≤ 0.05) in the control group with a mean (±SD) value of 0.04 (±0.04) from day 0 to day 14. On the other hand, in both the immunized groups, the RBA activity was significantly higher in comparison to the control over 2 weeks post-experiment. Among the immunized groups, the RBA activity was significantly high with a mean (±SD) value of 0.233 (±0.05) in the immersion group by day 7, which was constant by day 14 and then increased by day 32 and day 50 after the booster dose. A similar trend of RBA activity was seen in the bio-encapsulated group that had no significant difference (*p* ≥ 0.05) from the immersion group. The RBA activity of both groups increased by day 32 and was constant over day 50 by a value of 0.475 (±0.04.) The higher RBA activity was seen in the immersion group by day 50 at 0.55 (±0.06) (Figure 6).

#### 3.3.4. Serum Antiprotease Activity

The serum antiprotease activity measured as the percentage inhibition of trypsin activity in the control group was 25% over 2 weeks post-experiment. However, the percentage of inhibition in the immersion group was recorded at around 68%, which was not significantly different from the bio-encapsulated group, which showed 55% inhibition of trypsin on day 7 post-vaccination and around 49% on day 14. This activity was enhanced over day 32 and day 50, followed by a booster dose. On the other hand, significantly high inhibition (*p* ≤ 0.05) was observed in the groups immunized by immersion and bio-encapsulation than in the control group (Figure 7).

#### 3.3.5. Intestinal Tissue Myeloperoxidase (MPO) Activity

Intestinal tissue myeloperoxidase (MPO) activity was also significantly (*p* ≤ 0·05) enhanced in both immersion and bio-encapsulated groups as seen in serum MPO assay, in comparison with that of control on day 7 to day 50. There was also a significant difference (*p* ≤ 0·05) among the immersion and bio-encapsulated vaccination groups which shows increased MPO activity among the immersion route. The bio-encapsulation route exhibits higher and more constant MPO activity than the control group that persisted over day 14 to day 50. This activity is comparable to the MPO activity of the immersion group and is not significantly different (Figure 8).

#### 3.3.6. Alkaline Phosphatase (AKP)

Alkaline phosphatase (AKP) activity in the intestinal tissues of both immunization groups showed a significant increase by day 7 in comparison to the control group. In the bio-encapsulated group, the level of alkaline phosphatase was increased gradually after day 7 and was not significantly different to that of the immersion group. The AKP level of both immunized groups decreased by day 14, which increased by day 32 and day 50 with a booster dose and is not significantly different from each other. The highest AKP level was seen by day 32 with an OD value of 0.342 (±0.05) in the immersion group (Figure 9).

#### 3.3.7. Intestinal Tissue Lysozyme Activity

The intestinal tissue lysozyme activity showed increased activity of 3985 mg/mL at day 7 in the immersion group, which was constantly increasing over day 50 with a booster dose at day 30. The bio-encapsulated group showed a lysozyme activity of 3490 mg/mL on day 7, which was significantly higher than the control group. It increased by day 14 and was enhanced by a booster dose at day 32. Increased lysozyme activity was seen at day 50 by 6560 mg/mL and was not significantly different from the immersion group. The lysozyme activity of both the immunization groups was significantly different from the control group (Figure 10).

#### 3.3.8. Histology

Photomicrographs of the hindgut at 7 days post-experiment and 1-day post-challenge were recorded in all groups. Gut-associated lymphoid tissues (GALT) were observed in both immersion and bio-encapsulation groups. The induction of lymphoid cell aggregation in the lamina propria of immersion and bio-encapsulation groups was observed around one week post-immunization. No structures of GALT were developed in the lamina propria of the control group (H&E × 100). Post-challenge histological analysis of tissue samples was also carried out. Immersion and bio-encapsulation groups were seen with macrophages and lymphoid cells clustered over the villi and the villi length was seen as normal. Additionally, a normal arrangement of the mucosal epithelium (ME), submucosa (SM), muscularis (MM) and serosa was observed. Control group fishes showed epithelial necrosis with sloughed necrotic debris in their lumen and slight distortion in the microvilli structure (Figure 11). 

### 3.4. Challenge with Live Aeromonas hydrophila

#### Cumulative Mortality Index (CMI) and Post Challenge Symptoms

The mortality in intra-peritoneally challenged fishes started in all groups at 24 h post-challenge and levelled off between 48 to 96 h with an initial cumulative mortality index (CMI) of 50% (Figure 12) that led to total mortality in the control group fishes by 96 h. The immersion group fishes showed a CMI of 16.1% with a similar mortality pattern that was significantly different to that of the bio-encapsulated route of vaccination which showed a CMI of 26% over the challenge studies.

The challenged fishes were observed for symptoms associated with infection and recorded (Figure 13).

The relative percent survival rates of groups immersion, bio-encapsulation and control were 83.9%, 73.6% and 0%, respectively. The RPS of both the immunization groups were much higher than the RPS of the control group. The bio-encapsulation group ensures protection that is greater than the control group but less than the immersion group (Figure 14). The re-isolation of *A. hydrophila* from the challenged fishes after 3 weeks on Aeromonas Isolation Agar showed no growth for swabs collected from intestine, kidney and gill specimens from both immunized groups.

## 4. Discussion

As per the literature, several oral and immersion route immunizations were analyzed against *Aeromonas hydrophila* infection with varied results in aquaculture. Commercial oral and bath vaccines have been utilized in common carp cultivation with low effectiveness against *A. hydrophila,* corresponding to underlying factors [28]. This work primarily discusses the efficacy of oral vaccination in an economically significant freshwater fish *Cyprinus carpio* L., through live feed *Artemia salina*. In that aspect, the study carried out the assessment of the innate immune responses in *Cyprinus carpio* L. (common carp), upon the administration of heat-inactivated *Aeromonas hydrophila*, which is the causative agent of Motile Aeromonas Septicemia (MAS) disease in large-scale carp cultivation across the world. The study aimed to unveil the significance of bio-encapsulated antigenic delivery (earlier in practice) especially in common carp, against MAS infection, which is not reported yet in the literature. Thereby, it aims in assessing the role of this antigen to function as an inactivated vaccine through an enhanced route and dosage of administration, for a stronger innate immune response in different stages of carp. This could be commercialized for a large-scale fish culture and also adoptable for other commercial fish breeding systems. Here, we analyzed the efficacy of an oral *Artemia salina* bio-encapsulated bacterin immunization in common carp against *Aeromonas hydrophila*. The serum and hematological and intestinal tissue immune parameters were assessed along with tissue histological changes in a 60-day experiment. The effectiveness of this immunization technique was determined by assessing the relative percent survival followed by a lethal intraperitoneal *A. hydrophila* challenge.

The results of this investigation clearly indicate the innate immune responses in the *Cyprinus carpio* L. group immunized with bio-encapsulated bacterin were enhanced in comparison to those of the control group, and a significant difference was not observed between both the immersion and bio-encapsulated groups tested over 60 days of the experiment. Thus, these results support the concept of the inducible nature of the innate immune response in *Cyprinus carpio* L. upon administration with a bio-encapsulated bacterin that is discussed below in detail.

The optimized inactivation of *A. hydrophila* bacterin at 100 °C for 2 h was in accordance with the literature [29]. Following that, optimization and evaluation of the *A. hydrophila* bacterin at a concentration of 10^7^ CFU mL^−1^ uptake in *A. salina* nauplii was carried out. Our observation indicates that the *A. hydrophila* bacterin was completely bio-encapsulated by *Artemia salina* nauplii after 2 h of incubation [14] under appropriate conditions. The concentration of intake was correlated with a standard curve of *A. hydrophila* and found to be around 25 × 10^7^ CFU mL^−1^, and maximum retention in the gut was observed between the period of 0 to 120 min. The results were also verified with microscopic observations at different time intervals. Similar results were observed by [30], in which the bio-encapsulation of *V. anguillarum* bacterin showed maximum bio-encapsulation by the time period of 60 min to 120 min at a concentration of 1.5 × 10^7^ CFU mL^−1^. The bio-encapsulation in *A. salina* nauplii safeguards antigens from the digestive tract of the fish and promotes antigen detection by macrophages in the mucosal layer of the hindgut. Recent studies on artemia bio-encapsulated oral vaccine by [31] have also shown us the protection of antigen from gastrointestinal degradation and its smooth delivery to the hindgut for antigenic detection.

Myeloperoxidase (MPO), an immunological component of innate immune responses, is released from the phagocyte cytoplasmic granules that can contribute to oxidative responses against pathogenic invasion. The serum and intestinal tissue peroxidase levels were considerably (*p* ≤ 0.05) increased in the bio-encapsulated group from day 7 to day 14 experimental group fishes, and it was increased by a booster dosage at day 32. A similar study showed a significant increase in serum myeloperoxidase at an early infection in *P. sarana* following an intra-peritoneal challenge with *A. hydrophila* [32]. Thus, the higher peroxidase activity observed in bio-encapsulated and immersion groups not only portrays the stimulatory effect of bacterial factors on neutrophilic granulocytes, i.e., the inducible nature of peroxidases, but also indicates the role of the peroxidase–hydrogen peroxide system in the elimination of pathogenic factors.

Lysozyme, an innate immune defence enzyme, mediates protection against microbial invasion. Lysozyme is mostly found in the head kidney, leukocytes, gill, skin, gastrointestinal system and eggs, where bacterial invasion is quite likely. Two days after being infected with *A. hydrophila*, Atlantic salmon (*Salmo salar*) showed enhanced lysozyme activity [33]. The inducible nature of lysozyme activity reported earlier is again confirmed in the present investigation where the serum and tissue lysozyme response was found to be induced to a significantly higher level in comparison to the control and immersion groups in fishes from day 14 to day 50.

Similarly, serum antiprotease enzyme also actively engages in the innate immune responses of fish. Several bacterial pathogens invade the host and obtain their nutrients by using their extracellular proteolytic enzymes. To neutralize such pathogenic proteases, the host organism uses various protease inhibitors such as α1-antiproteinase and α2-macroglobulin (α2M) present in the serum and other body fluids which have anti-enzyme activity [34]. These protease inhibitors could selectively arrest the replication of bacterial pathogens without untoward toxicity to the host. Inhibition of trypsin activity is the most convenient way of measuring the anti-protease activity of the serum. Our study indicates that the antiprotease level in common carp on all the post-experimental days tested was significantly elevated in the bio-encapsulated group than the control but was lesser in comparison to the immersion due to the concentration of bacterin entering the fish during immunization. The results indicate that antiprotease neutralizes the bacterial proteases and thus gradually gets rid of the infection. This discovery agrees with the literature that in *C. striata* [35] experimentally infected with *A. hydrophila*, the expression of Kazal-type antiprotease was greater than in control (uninfected) fish [36].

Respiratory burst activity (RBA) is the rapid release of reactive oxygen species (O_2_ and H_2_O_2_) from different types of cells. Usually, it denotes the release of these chemicals from immune cells, e.g., neutrophils and monocytes, as they come into contact with different pathogens. A previous study showed that injection of pathogenic *A. hydrophila* into *C. striata* [35] caused the up-regulation of tumor necrosis factor α (TNF-α) within 24 h, which subsequently led to ROS production by phagocytic cells. In that aspect, an enhanced level of blood RBA level shown in the bio-encapsulated group compared to the control group among juveniles indicates the interaction of antigen with intestinal and serum immune cells such as neutrophils and monocytes.

In another work, the effect of *Aeromonas hydrophila* infection on the non-specific immunity of blunt snout bream (*Megalobrama amblycephala*) was studied. It showed that after the challenge, there was an increase in the alkaline phosphatase activity [37]. Alkaline phosphatase activity in the treatment group showed a significant increase at 4 h, 1 d, 3 d, 5 d, 14 d and 21 d compared to the control group. Increased phosphatase activity indicates a higher breakdown of the energy reserve, which is utilized for the growth and survival of fish. These results revealed that the non-specific immunity of fish played an important role in self-protection after pathogenic infection. This is in correlation with the obtained results which showed an increased level of AKP activity in both the treatment fishes in comparison to the control group.

The presence of Gut Associated Lymphoid Tissues (GALT) and lymphoid cell aggregation in immunized fish in comparison to the control fish over 7 days after the experiment and the necrosis occurrence only in control fish at post-challenge were in relation to the results obtained from previous studies on histopathological effects of Aeromonas infection. One such study shows the hemocyte aggregation in the digestive system of blue tilapia, *Oreochromis aureus,* upon *A. hydrophila* infection [38].

The mortality and relative percent survival analysis for evaluating the protection showed that the cumulative mortality index (CMI) in the bio-encapsulated group was much lower than the control group but lesser in comparison to the immersion route of immunization. After an intraperitoneal challenge, the bio-encapsulated group showed a decreased cumulative mortality index (CMI) than the control group, which ensures protection in common carp. With continuous booster dosage, it also showed stable protection in comparison to the immersion group. These results were in relation to the previous studies with artemia bio-encapsulated vaccination of juvenile carp of 15, 29 and 58 days old with a *Vibrio anguillarum* bacterin [8]. The RPS analysis in the present work showed that in treated fish, protection was at an RPS of 73% and 84% in both bio-encapsulated and immersion routes, respectively. Similar results were obtained in an analysis with lumpfish orally immunized as larvae and then both orally and i.p. boosted as juveniles that showed a significant RPS (76.5%) to the *V. anguillarum* i.p. challenge [31]. This suggests that the orally administered bio-encapsulated immunization confers protection to the fish by minimizing handling stress and can be aided with an enhanced booster dose for stronger protection. The re-isolation of *A. hydrophila* from the surviving fishes after 4 weeks of challenge also showed the absence of bacteria in the immunized group.

In conclusion, the present study analyzed the non-specific immune response parameters in *Cyprinus carpio* L., which is shown to be sensitive and inducible when administered with bio-encapsulated *Aeromonas hyrophila* antigen. These findings point to a feasible method by which *A. hydrophila* may induce immunological modulation and contribute to a better understanding of the mechanism of action of inactivated vaccines in fish. Hence, this method could be widely applicable in large-scale aquaculture, utilizing *Artemia salina* as a bio-carrier for recombinant vaccines and probiotics production. Serum and cellular immune response parameters studied herein can be used to assess the health analysis of common carp upon *A. hydrophila* infection. Future studies should focus on the cellular and molecular events taking place during *A. hydrophila* infection in *Cyprinus carpio* L. at transcriptomic and proteomic levels, thereby formulating better dosage and immunization techniques in large-scale aquaculture.

## 5. Conclusions

In conclusion, the present study analyzed the non-specific immune response parameters in *Cyprinus carpio* L., which is shown to be sensitive and inducible when administered with bio-encapsulated *Aeromonas hyrophila* antigen. These findings point to a feasible method by which *A. hydrophila* may induce immunological modulation and contribute to a better understanding of the mechanism of action of inactivated vaccines in fish.

In comparison with other immunization methods, the oral route imposes less stress, healthy immune response stimulation and easier administration in mass aquaculture along with cost efficiency. Live feeds such as *Artemia salina* can be employed in such cases as potent biological carriers owing to their smaller size, easier cultivation as larval feed and non-filter-feeding characteristics to carry a wider range of bacterin as potent vaccines. Hence, this work aimed at assessing the innate immune response of an oral immunization of *Cyprinus carpio* L. juveniles using *Artemia salina* bio-encapsulated heat-killed *A. hydrophila* against Motile Aeromonas Septicemia (MAS) disease. Results conferred that bacterin stimulated the innate immune response in juvenile-stage *Cyprinus carpio* L. The enhanced innate immune responses such as differential leucocyte count (DLC), lysozyme, respiratory burst activity, myeloperoxidase activity, serum antiprotease activity, and alkaline phosphatase activity of serum, blood and intestinal tissue samples were seen in juvenile fishes immunized with a bio-encapsulated antigen. All modulation of innate immune parameters of juveniles analyzed at intervals of the experiment was significantly different than the control group but less and comparable to the immersion group fishes. The challenge study showed reduced mortality over the control group and the relative percent survival was more significant in the treatment group. Hence, this method could be widely applicable in large-scale aquaculture, utilizing *Artemia salina* as a bio-carrier for recombinant vaccines and probiotics production. Serum and cellular immune response parameters studied herein can be used to assess the health analysis of common carp upon *A. hydrophila* infection. Future studies should focus on the cellular and molecular events taking place during *A. hydrophila* infection in *Cyprinus carpio* L. at transcriptomic and proteomic levels, thereby formulating better dosage and immunization techniques in large-scale aquaculture. Vaccine administration through these carrier systems via the oral route in live feeds will be extremely useful for the prevention and control of *Aeromonas hydrophila* infections, which are more prevalent in common carp (*Cyprinus carpio* L.) and can also be applicable to other aquatic diseases in the future.

## Figures and Tables

**Figure 1 vaccines-11-00877-f001:**
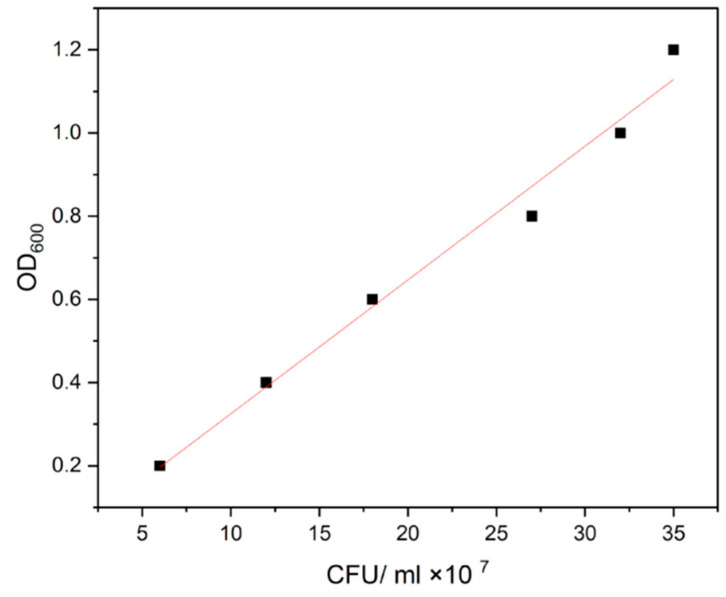
Standard curve of *Aeromonas hydrophila* to determine the concentration of bacterin uptaken by Artemia nauplii (R^2^ = 0.98875; y = 0.3848x − 0.0805).

**Figure 2 vaccines-11-00877-f002:**
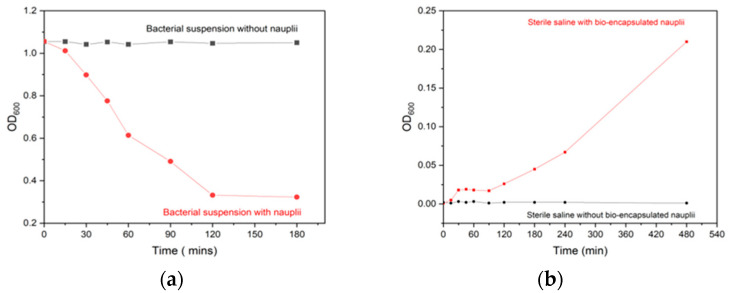
(**a**). Optimization of time of maximum uptake of heat-killed *A. hydrophila* bacterin (10^7^ CFU mL^−1^) in Artemia nauplii. (**b**) The retention time of heat-killed *A. hydrophila* bacterin (10^7^ CFU mL^−1^) in Artemia nauplii was analyzed for up to 4 h.

**Figure 3 vaccines-11-00877-f003:**
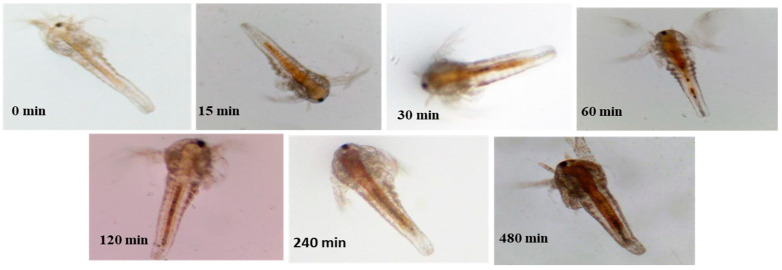
Microscopic observation of Artemia nauplii during bio-encapsulation and retention analysis.

**Figure 4 vaccines-11-00877-f004:**
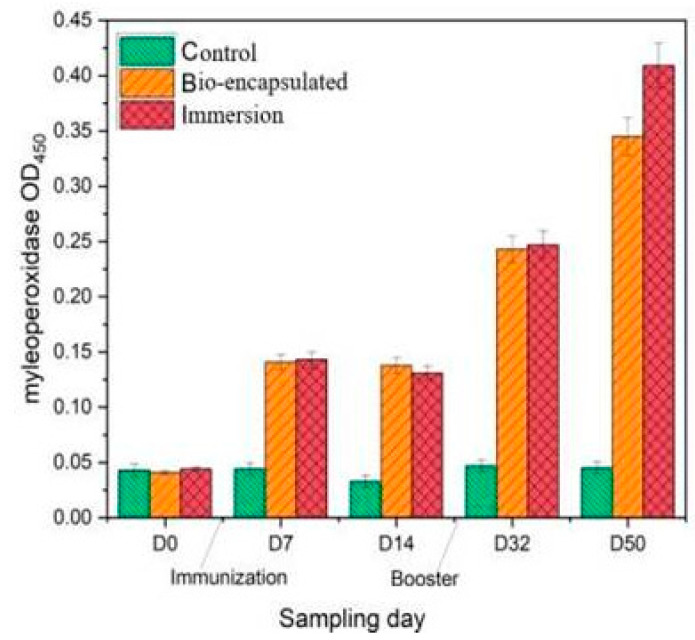
Serum myeloperoxidase (MPO) activity of the juvenile *Cyprinus carpio* L. in all three groups, control, bio-encapsulated and immersion at day 0, 7, 15, 32 and 50.

**Figure 5 vaccines-11-00877-f005:**
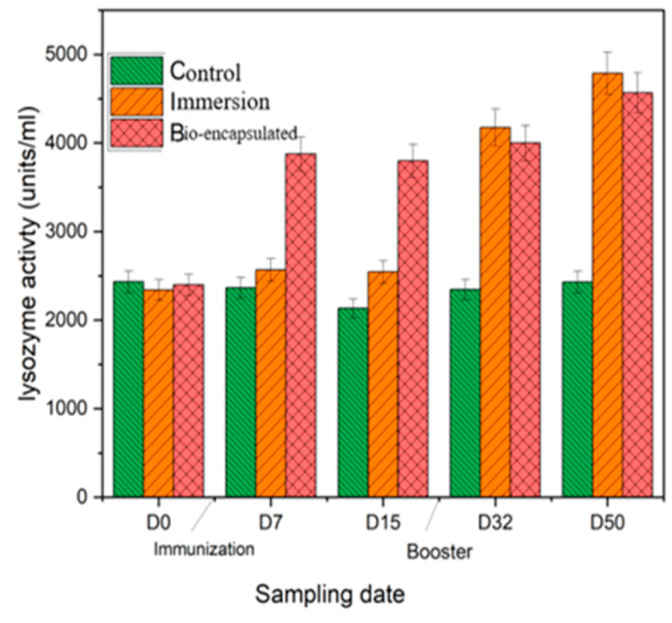
Serum lysozyme activity of the juvenile *Cyprinus carpio* L. in all three groups, control, bio-encapsulated and immersion at day 0, 7, 15, 32 and 50.

**Figure 6 vaccines-11-00877-f006:**
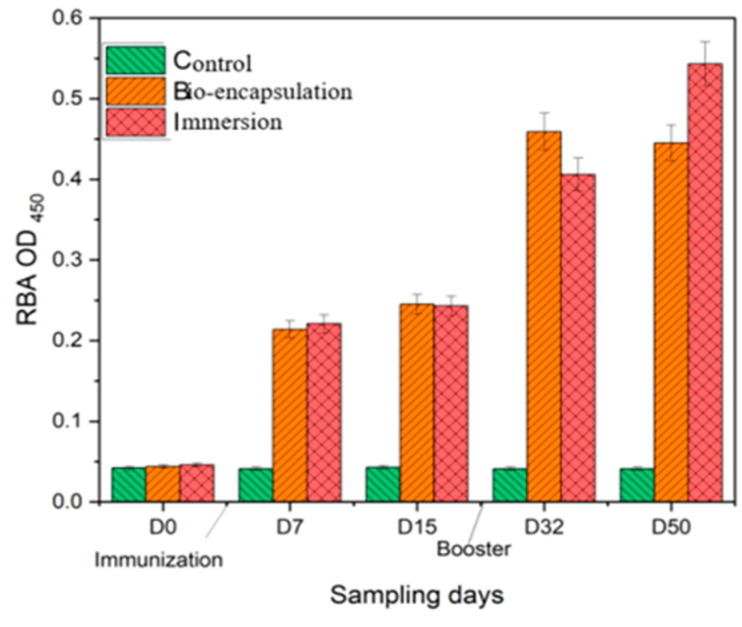
Respiratory Burst Activity of the juvenile *Cyprinus carpio* L. in all three groups, control, bio-encapsulated and immersion at day 0, 7, 15, 32 and 50.

**Figure 7 vaccines-11-00877-f007:**
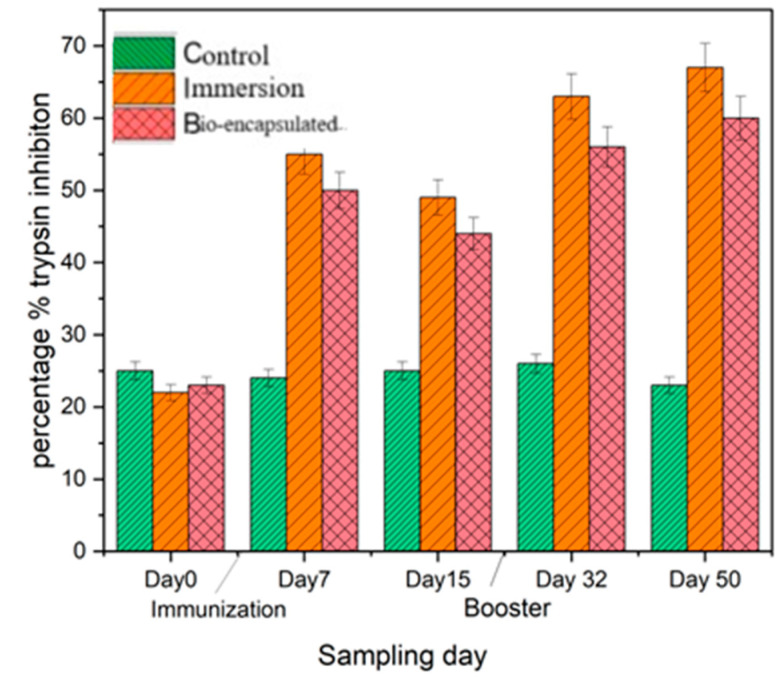
Serum antiprotease activity of the juvenile *Cyprinus carpio* L. in all three groups, control, bio-encapsulated and immersion at day 0, 7, 15, 32 and 50.

**Figure 8 vaccines-11-00877-f008:**
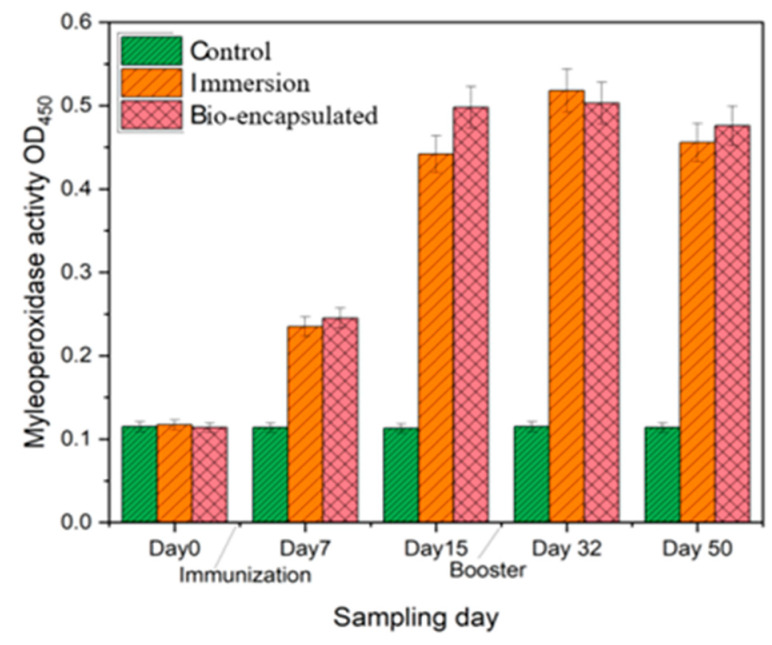
Intestinal tissue myeloperoxidase activity (MPO) of the juvenile *Cyprinus carpio* L. in three groups, control, bio-encapsulated and immersion at day 0, 7, 15, 32 and 50.

**Figure 9 vaccines-11-00877-f009:**
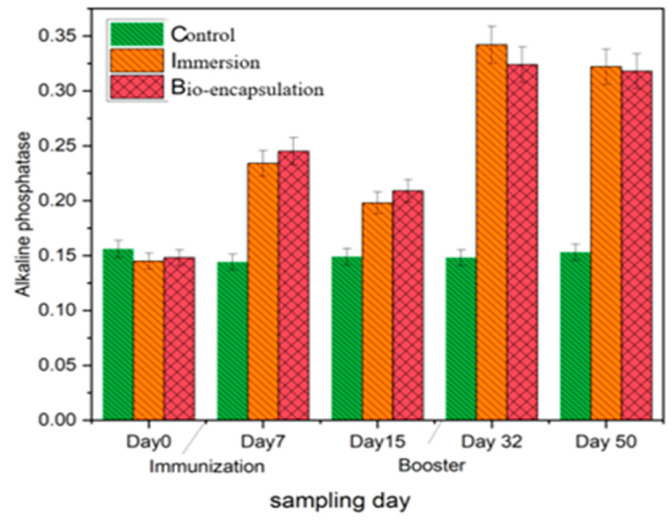
Intestinal tissue alkaline phosphatase activity of the juvenile *Cyprinus carpio* L. in three groups, control, bio-encapsulated and immersion at day 0, 7, 15, 32 and 50.

**Figure 10 vaccines-11-00877-f010:**
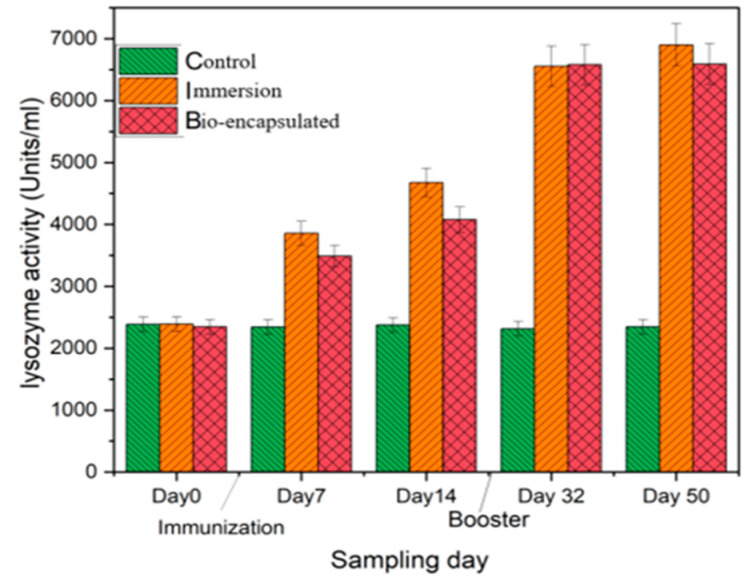
Intestinal tissue lysozyme activity of the juvenile *Cyprinus carpio* L. in three groups, control, bio-encapsulated and immersion at day 0, 7, 14, 32 and 50.

**Figure 11 vaccines-11-00877-f011:**
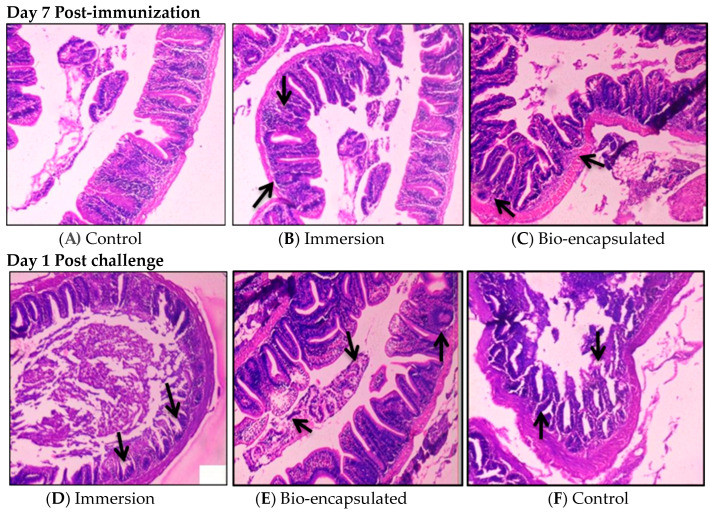
Histological studies of hindgut tissue in all groups of juvenile fishes (Control, Immersion and Bio-encapsulation) at 7 days post-immunization and 1 day post-challenge. (**B**,**C**)-GALT tissue formation, lymphoid cell aggregation. No difference in control (**A**) at 7 d.p.v and at 1.d.p.c. (**D**,**E**)—lymphoid cell aggregation, presence of macrophages. In the control group (**F**), necrosis and distortion in the microvilli structure was seen.

**Figure 12 vaccines-11-00877-f012:**
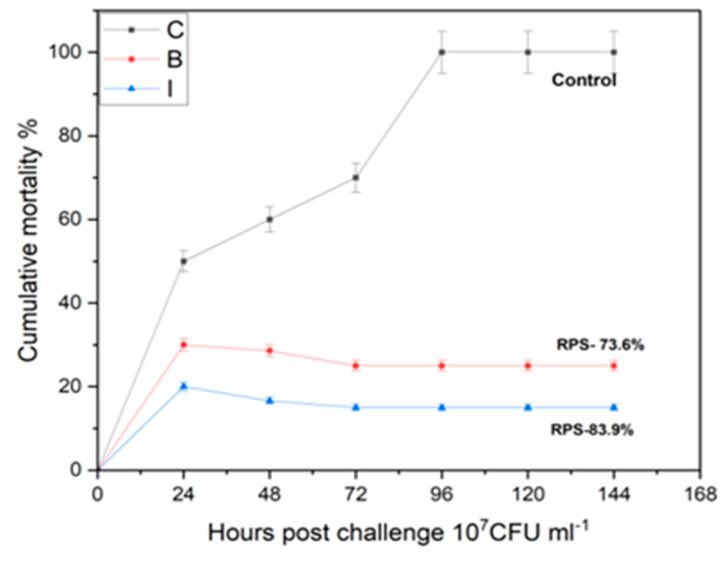
Cumulative mortality index (CMI) of all three groups (Control, Immersion and Bio-encapsulated) postchallenge.

**Figure 13 vaccines-11-00877-f013:**
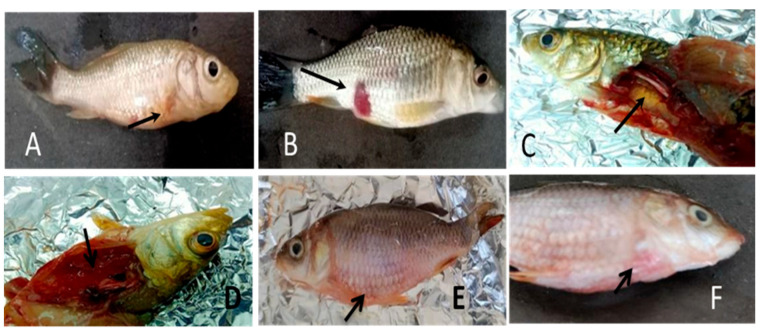
Symptoms post-challenge in fish prior to mortality (**A**,**E**,**F**) External hemorrhage, Reddened fins, Iridial hemorrhage, (**E**) Swollen abdomen, Detachment of Scales (**B**) Open bloody sores/ulcers, exophthalmia. (**D**) Internal hemorrhage/Ulcerative necrosis (**C**) Friable liver, and congested kidney with serous fluid.

**Figure 14 vaccines-11-00877-f014:**
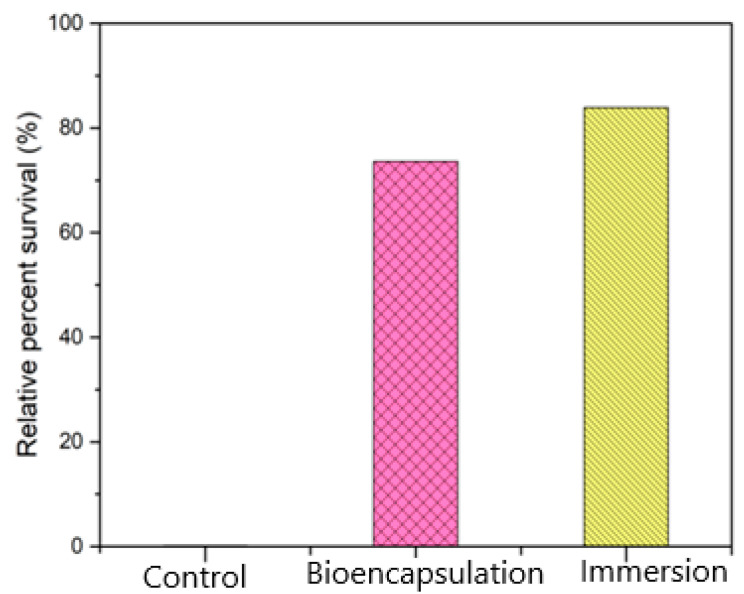
Relative Percent Survival (RPS) of all three groups (Control, Immersion & Bio-encapsulation) post-challenge.

**Table 1 vaccines-11-00877-t001:** Different leukocyte count of blood samples of all three group fishes (n = 3) at 14 days post-vaccination.

Types of Cells	Control	Immersion	Bioencapsulation
Lymphocytes	85 ± 1.0	82.6 ± 1.15	82.6 ± 2.08
Monocytes	4.6 ± 0.5	4.6 ± 1.15	3.6 ± 1.5
Granulocytes	11 ± 1.0	12.6 ± 2.0	12 ± 2.6

## Data Availability

The data for this study is available in this article.

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
