# Peer review of "Innate Immune Response Assessment in Cyprinus carpio L. upon Experimental Administration with Artemia salina Bio-Encapsulated Aeromonas hydrophila Bacterin"

_vaccines, 2023, doi:10.3390/vaccines11040877_

Round 1

Reviewer 1 Report

L571-574

Although the effectiveness of the Bio-encapsulation system was demonstrated in this study, unfortunately, it was less effective than the immersion method. The authors may believe that there are advantages of this system over the immersion method other than vaccine efficacy, but these advantages are not clearly presented for the reader to understand. Some advantage should be clearly stated. The author mentions that one of the advantages of the oral method is that it is less stressful, which is certainly true when compared to the injection method, but is it not less stressful than the immersion method? Also, the other benefits, ease of administration and cost, are not stated in a way that the reader can understand how easy it is to administer and how much less costly it is compared to the immersion method. As for ease of administration, I believe the immersion method is easier. If this is not clearly stated, the advantages of this method will not be conveyed at all.

Fig. 4 and 6

The coloring of the legend differs from the others (Figures 5, 7. 8, 9, 10); Bio-encapsulated should be corrected to red and Immersion to orange to be consistent. The order of the graphs should also be consistent.

Author Response

Reviewer 1

Comments and Suggestions for Authors   L571-574

Although the effectiveness of the Bio-encapsulation system was demonstrated in this study, unfortunately, it was less effective than the immersion method. The authors may believe that there are advantages of this system over the immersion method other than vaccine efficacy, but these advantages are not clearly presented for the reader to understand. Some advantage should be clearly stated. The author mentions that one of the advantages of the oral method is that it is less stressful, which is certainly true when compared to the injection method, but is it not less stressful than the immersion method? Also, the other benefits, ease of administration and cost, are not stated in a way that the reader can understand how easy it is to administer and how much less costly it is compared to the immersion method. As for ease of administration, I believe the immersion method is easier. If this is not clearly stated, the advantages of this method will not be conveyed at all.

Corrected the statements regarding the advantage of bio-encapsulation method over immersion & injection route.

Fig. 4 and 6

The coloring of the legend differs from the others (Figures 5, 7. 8, 9, 10); Bio-encapsulated should be corrected to red and Immersion to orange to be consistent. The order of the graphs should also be consistent.

The figure 4 & 6 has been changed as per the query.

Reviewer 2 Report

This study explored innate immune response assessment in Cyprinus carpio L. upon experimental administration with Artemia salina bio-encapsulated Aeromonas hydrophila bacterin. The topic of this study is practical important. However, this study complicates a simple problem and has little effect on production. As far as I know, common carp can accept artificial compound feed when their body length is more than 1cm. Therefore, applying live feeds has no practical application value and will greatly increase the production cost.

Line 31: The first letter of a word should be lowercase.

Lines 32-33: change “Most of the non-specific (both humoral and cellular)” to “Both humoral and cellular”

Line 47: change “seawater, or freshwater,” to “seawater or freshwater,”

Line 81: “(US$ 3-5 per kg)”, is this the international market price?

Line 93: delete “90;”

Line 100: please provide the body length of fish upon the experiment was conducted.

Lines 100-101: please rewrite this sentence “After 4 weeks of acclimation, the fish were transferred to three experimental tanks in duplicate (15 fish per tank). ”, such description will mislead the readers.

Line 111: why n=2?

Author Response

Reviewer 2

Comments and Suggestions for Authors

This study explored innate immune response assessment in Cyprinus carpio L. upon experimental administration with Artemia salina bio-encapsulated Aeromonas hydrophila bacterin. The topic of this study is practical important. However, this study complicates a simple problem and has little effect on production. As far as I know, common carp can accept artificial compound feed when their body length is more than 1cm. Therefore, applying live feeds has no practical application value and will greatly increase the production cost.

Line 31: The first letter of a word should be lowercase.

         Corrected

Lines 32-33: change “Most of the non-specific (both humoral and cellular)” to “Both humoral and cellular”

         Sentence Corrected to “Both humoral and cellular”

Line 47: change “seawater, or freshwater,” to “seawater or freshwater,”

         Changed “seawater, or freshwater,” to “seawater or freshwater,”

Line 81: “(US$ 3-5 per kg)”, is this the international market price?

Line 81: “(US$ 3-5 per kg)”this is the international market price.

Line 93: delete “90;”

         Line 93: deleted “90;”

Line 100: please provide the body length of fish upon the experiment was conducted.

         Added, length of the fish subjected to experiment.

Lines 100-101: please rewrite this sentence “After 4 weeks of acclimation, the fish were transferred to three experimental tanks in duplicate (15 fish per tank).”, such description will mislead the readers.

         Lines 100-101: Rewrote the sentence.

Line 111: why n=2?

n=2 from each experimental setup. Duplicates were also maintained. Since samples are collected at different intervals multiple time and also considering the total number of fishes.

Reviewer 3 Report

Generally speaking, this study reported a valid method for the inactivated vaccine in fish by administering with bio-encapsulated A. hysrophila antigen. The experiment was well-designed and the results are convincing. However, a major revision is needed before its acceptance for publication. Specific comments are as follows:

1.      Although you reported some significant differences between treatments, you did not show them in the tables and figure. Please add the significance labels in all of the figures and tables.

2.      Line 111, you said n=2 here, please check whether it is right.

3.      Line 246, you used student’s t-test. In my opinion, you should use tukey or SNK q-test, not t-test, to avoid the inconsistent se. Please redo the post-hoc comparisons.

4.      Line 289 with table 1: the significant figures for Lymphocytes are different, please revise this.

5.      Figure 11 and 13, Please use different symbols to show the different symptoms in the pictures.

6.      Figure 12, RPS-73.6% and RPS-83.9% seem strange, please delete them.

7.      Line 422-428: I think this paragraph should not separate from 3.4.1. Essentially, you are describing the mortality or survival. And you also should remove the Figure 14, as it is a duplicate of part of results from Figure 13. As you used RPS in the method, so I also suggest you to use RPS in this section, rather than CMI.

8.      Line 542-552, in this paragraph, you drew a conclusion. However, in section 5, you also have a conclusions section. I think you should integrate this paragraph into section 5.

9.      References: All the scientific names for species should be in italic.

Author Response

Reviewer 3

Comments and Suggestions for Authors

Generally speaking, this study reported a valid method for the inactivated vaccine in fish by administering with bio-encapsulated A. hysrophila antigen. The experiment was well-designed and the results are convincing. However, a major revision is needed before its acceptance for publication. Specific comments are as follows:

  1. Although you reported some significant differences between treatments, you did not show them in the tables and Please add the significance labels in all of the figures and tables.

         Significant difference has been mentioned through error bars in all graphs.

  1. Line 111, you said n=2 here, please check whether it is

        Yes, n=2 is correct considering thetotal number of experimental fishes since samples were collected at different intervals.

  1. Line 246, you used student’s t-test. In my opinion, you should use tukey or SNK q-test, not t-test, to avoid the inconsistent Please redo the post-hoc comparisons.

         T-test adopted for significant comparison.

  1. Line 289 with table 1: the significant figures for Lymphocytes are different, please revise

        Revised.

  1. Figure 11 and 13, Please use different symbols to show the different symptoms in the

Different symptoms revised with new symbols.

  1. Figure 12, RPS-73.6% and RPS-83.9% seem strange, please delete

        RPS percentage removed from Fig.12

  1. Line 422-428: I think this paragraph should not separate from 4.1. Essentially, you are describing the mortality or survival. And you also should remove the Figure 14, as it is a duplicate of part of results from Figure 13. As you used RPS in the method, so I also suggest you to use RPS in this section, rather than CMI.

        Changes made accordingly.

  1. Line 542-552, in this paragraph, you drew a However, in section 5, you also have a conclusions section. I think you should integrate this paragraph into section 5.

        The respective paragraph integrated in Sec 5

  1. References: All the scientific names for species should be in

Every scientific names were italicized in the Reference section.

Round 2

Reviewer 2 Report

The implications of the study are not addressed by the authors.

Author Response

This study explored innate immune response assessment in Cyprinus carpio L. upon experimental administration with Artemia salina bio-encapsulated Aeromonas hydrophila bacterin. The topic of this study is practical  important. However, this study complicates a simple problem and has little effect on production. As far as I know, common carp can accept artificial compound feed when their body length is more than 1 cm. Therefore, applying live feeds has no practical application value and will greatly increase the production cost.

The aim of this study was to compare the efficacy of  inactivated A.hydrophilla vaccine through live feed Artemia that could minimize the immunization stress that exists in both injection and immersion route. Live feeds such as Artemia salina can be employed in such cases as potent biological carriers owing to their smaller size, easier cultivation as larval feed and non-filter feeding characteristics to carry a wider range of bacterin as potent vaccines. Previous studies have shown that live feeds could act as immune stimulating adjuvants as a carrier  that could correspond to the growth of the host being administered.

Also in terms of  artificial compound feeds, there are chances for the compound to leach out and degrade that can be inefficient in terms of vaccination . In this method, we have optimized the appropriate time and dosage of bio-encapsulation and vaccination that could follow  the natural route of intake in fishes of both sizes. The live feed could also support the nutrition of fishes during their immunization process. Considering the ease of handling in comparison to injection methods in a mass cultivation and  large productivity costs for preparing formulated artificial compound feed , this method utilizing live feeds will be cost effective and worthwhile.

Line 31: The first letter of a word should be lowercase.

         Corrected

Lines 32-33: change “Most of the non-specific (both humoral and cellular)” to “Both humoral and cellular”

         Sentence Corrected to “Both humoral and cellular”

Line 47: change “seawater, or freshwater,” to “seawater or freshwater,”

         Changed “seawater, or freshwater,” to “seawater or freshwater,”

Line 81: “(US$ 3-5 per kg)”, is this the international market price?

Line 81: “(US$ 3-5 per kg)”this is the international market price.

Line 93: delete “90;”

         Line 93: deleted “90;”

Line 100: please provide the body length of fish upon the experiment was conducted.

         Added,length of the fish subjected to experiment.

Lines 100-101: please rewrite this sentence “After 4 weeks of acclimation, the fish were transferred to three experimental tanks in duplicate (15 fish per tank).”, such description will mislead the readers.

         Lines 100-101: Rewrote the sentence.

Line 111: why n=2?

n=2 from each experimental setup. Duplicates were also maintained. Since samples are collected at different intervals multiple times and also considering the total number of fishes.

Reviewer 3 Report

I think the authors did not revise the manuscript according to my comments except my comment 9.  

Author Response

(The authors gave the same response as above.)

Round 3

Reviewer 2 Report

The authors have addressed all the comments.

Author Response

Comments and Suggestions for Authors The authors have addressed all the comments.   We thank the reviewers for their valuable time and comments in improving the manuscript.